

# BMP2 alterations in mucinous cystadenocarcinoma of the breast: insights from whole-exome sequencing

Zhenyu Li[1,2,*], Yi Gong[3,*], Guangxin Li[1], Qingming Jiang[1,4], Juanhui Dong[1,4] and Rui Chen[1,2]

[1] Department of Pathology, Chongqing University Cancer Hospital, Chongqing, China
[2] Chongqing Key Laboratory for the Mechanism and Intervention of Cancer Metastasis, Chongqing University Cancer Hospital, Chongqing, China
[3] Department of Phase I Clinical Trial Ward, Chongqing University Cancer Hospital, Chongqing, China
[4] Chongqing Cancer Multi-omics Big Data Application Engineering Research Center, Chongqing University Cancer Hospital, Chongqing, China
* These authors contributed equally to this work.

Corresponding authors
Juanhui Dong,
Dongjuanhui1984@163.com
Rui Chen, chenrui76@cqu.edu.cn

## ABSTRACT

**Background:** Mucinous cystadenocarcinoma (MCA) of the breast is a rare and distinct type of mammary tract adenocarcinoma. Molecular alterations in breast MCA remain poorly characterized in current epidemiological data.
**Methods:** We summarized the clinicopathological characteristics and performed exome sequencing on formalin-fixed, paraffin-embedded blocks from three patients with mammary MCA culled from the surgical pathology archives of Chongqing University Cancer Hospital. The tumor mutation landscape of these patients was determined using whole-exome sequencing. We assessed cancer effect sizes and analyzed the contribution of mutational processes to each oncogenic variant, quantifying the extent to which each process contributed to tumorigenesis/cancer effects.
**Results:** Our analysis revealed that one of the three patient cohorts with advanced-stage breast MCA demonstrated aggressive clinical behavior, which contrasts with the typically reported indolent course of this rare malignancy. We also identified a significant mutant gene, bone morphogenetic protein 2 (BMP2), that is associated with breast MCA. Furthermore, we evaluated the expression by immunohistochemistry in patients with breast MCA and predicted the potential molecular functions of BMP2 in breast cancer through *in silico* analyses.
**Conclusion:** This study provides new insights into the clinical and molecular features of breast MCA, suggesting that tumor stage is an important prognostic indicator. BMP2 emerged as a potential driver of tumor progression, and its utility as a predictive or prognostic biomarker warrants further investigation in larger cohorts. These findings may pave the way for improved prognostic assessment and novel therapeutic strategies targeting BMP2 in breast MCA.

---

## INTRODUCTION

Breast cancer is the most common malignancy in females and the leading cause of cancer-related mortality in women worldwide, placing immense pressure on public health resources (*Arnold et al., 2022*; *Bray et al., 2024*). Mucinous cystadenocarcinoma (MCA) of the breast represents an extraordinarily rare histological variant, with fewer than 50 reported cases to date in English literature (*Chen et al., 2004*, *2023*; *Honma et al., 2003*; *Kim et al., 2012*; *Koenig & Tavassoli, 1998*; *Koufopoulos et al., 2017*; *Kucukzeybek et al., 2014*; *Lee & Chaung, 2008*; *Li et al., 2012*; *Nayak et al., 2018*; *Petersson et al., 2010*; *Seong et al., 2016*; *Sun et al., 2020*; *Vegni et al., 2023*; *Wang et al., 2020*). MCAs in the breast histologically resemble pancreatobiliary or ovarian MCAs, characterized by cystic structures lined with tall columnar mucinous cells, with invasive stratification, tufting, or complex papillary architecture and present an affecting predisposition in post-menopausal Asian women (*Joneja & Palazzo, 2023*).

Intriguingly, although most MCAs of the breast exhibit immunohistochemical negativity for estrogen receptor (ER), progesterone receptor (PR), and human epidermal growth factor receptor 2 (HER-2), they generally present with a significantly more favorable prognosis compared to conventional basal-like triple-negative breast cancers (*Joneja & Palazzo, 2023*; *Lei, Shi & Chen, 2023*; *Moatasim & Mamoon, 2022*). This prognostic discrepancy underscores the need to elucidate the underlying molecular mechanisms and tumor-microenvironment interactions that might confer this prognostic advantage.

Bone morphogenetic protein 2 (BMP2), a member of the transforming growth factor-β (TGF-β) superfamily, has emerged as a critical regulator participating in many developmental processes, including cardiogenesis, neurogenesis, and osteogenesis (*Lavery et al., 2008*; *Seeherman et al., 2019*; *Tanaka et al., 2014*). While originally discovered for its osteoinductive properties and clinical utility to enhance bone repair and regeneration, BMP2 has demonstrated complex, context-dependent roles in tumorigenesis (*James et al., 2016*). Recent evidence suggests that alterations in bone morphogenetic protein (BMP) signaling may influence breast cancer progression and metastasis (*Liu et al., 2023*); however, the specific function and regulatory targets of BMP2 in breast MCA pathogenesis are yet to be fully understood.

In recent years, whole-exome sequencing (WES) has emerged as a valuable tool for investigating the molecular basis of tumors including rare cancers, through comprehensive analysis of all protein-coding regions within the genome, thus providing powerful support for precision oncology (*Ganatra et al., 2024*). While the genetic landscapes of common breast cancer subtypes have been well characterized in the era of precision medicine, studies focusing on the molecular alterations in MCA of the breast remain limited. previous studies have reported HER2-positivity/amplification was reported in four MCA cases (*Kaur et al., 2022*; *Kucukzeybek et al., 2014*; *Petersson et al., 2010*; *Seong et al., 2016*). Additionally, targeted next generation sequencing (NGS) using a 425–580 cancer-related gene panel or WES has identified recurrent mutations in PIK3CA (phosphatidylinositol-4,5-bisphosphate 3-kinase catalytic subunit alpha), KRAS (kirsten rat sarcoma viral

oncogene homolog), MAP2K4 (mitogen-activated protein kinase kinase 4), RB1 (Retinoblastoma 1), KDR (kinase insert domain receptor), PKHD1 (polycystic kidney and hepatic disease 1), TERT (telomerase reverse transcriptase), TP53 (tumor protein p53) (*Chen et al., 2024*; *Lei, Shi & Chen, 2023*; *Lin et al., 2021*). Nevertheless, the overall genomic profile of mucinous cystadenocarcinoma (MCA) of the breast and its implications for oncogenesis and clinical phenotype remain largely elusive, necessitating more in-depth exploration.

To address this knowledge gap, we performed whole-exome sequencing of genomic DNA (deoxyribonucleic acid) from paired tumor and adjacent noncancerous breast tissue in three MCA patients. The mutational profile of breast MCA and the correlation between molecular alterations and clinicopathological characteristics were systematically analyzed. Our study aims to identify key driver mutations and potentially targetable pathways that might contribute to the unique biological behavior of this rare breast cancer subtype, offering insights that could enhance our understanding of breast MCA pathogenesis and potentially guide the development of tailored therapeutic strategies.

## MATERIALS AND METHODS

### Tissue collection

This study was conducted in accordance with the Declaration of Helsinki and approved by the ethics committee of Chongqing University Cancer Hospital (approval number: CZLS2023062-A). The ethics committee granted a waiver of informed consent for this retrospective study, while biological sample storage informed consent was maintained. Cases were selected based on the following criteria: histopathological diagnosis of breast mucinous cystadenocarcinoma, tumor tissue containing >50% tumor cells, and availability of complete clinical data. Cases with insufficient tissue material, poor DNA quality, or incomplete clinical information were excluded.

Based on these criteria, we retrospectively identified three female patients diagnosed with breast MCA who were treated at Chongqing University Cancer Hospital between June 2020 and October 2022. Formalin-fixed, paraffin-embedded (FFPE) tissue samples were collected from these patients. Relevant clinical and pathological data were extracted from medical records and are summarized in Table 1.

### Hematoxylin and eosin (H&E) staining and whole-slide image acquisition

The representative FFPE tissue blocks of breast tumor from the patients were sectioned at 5 μm for H&E staining (VENTANA HE 600 system; Roche Diagnostics, Indianapolis, IN, USA). All the stained slides were scanned at 20× magnification, with an image resolution of 0.25 mm/pixel scanner (MAGSCANNER, KF-PRO-005-HI; KF Biopathology, Ningbo City, China).

### Whole exome sequencing and bioinformatics analysis

DNA extraction and whole-exome Next-Generation sequencing were performed in the laboratory of Beijing Tsingke Biotech Co., Ltd. according to standard procedures. Briefly,

**Table 1  Characteristics of the patients.**

| Parameters | Case 1 | Case 2 | Case 3 |
|---|---|---|---|
| Age | 53 | 66 | 55 |
| Location (maximal diameter) | Six o'clock in left breast (4.0 cm); Outer quadrants of the right breast (15.0 cm) | Upper-Outer quadrants of the right breast (3.5 cm) | Upper-Inner quadrants of the left breast (3.0 cm) |
| Pathological diagnosis | Left breast: IDC, NOS, grade 3 (Nottingham Grade), G1 (Miller-payne Grade); Right breast: BMCA, grade 2 (Nottingham Grade), G1 (Miller-payne Grade). With metastasis to the skin of the right chest wall | BMCA (60%) with IDC, NOS, grade 2 (Nottingham Grade). | BMCA, grade 3 (Nottingham Grade). |
| Immune phenotype | IDC, NOS: ER (−), PR (−), EGFR (+), HER-2 (−), Ki-67 (60%); BMCA: ER (−), PR (−), HER-2 (−), EGFR (+) Ki-67 (60%). | ER (−), PR (−), HER-2 (−), Ki-67 (40%). | ER (−), PR (−), HER-2 (1+), Ki-67 (40%). |
| Stage | Left breast cancer: ypT4N2M1; rT4N2M1 stage IV; Right breast cancer: ypT4N2M1; rT4N2M1 stage IV. | Right breast cancer: pT2N0M0 stage IIA. | Left breast cancer: pT2N0M0 stage IIA. |
| Genetics | No germline PV was detected. See additional NGS results. | See NGS results. | See NGS results. |
| Treatment | NACT with TAC*7 cycles & NP*3 cycles; followed by extended radical mastectomy for left breast cancer and right breast cancer subsequentially; post-operative chemotherapy with NX*7 cycles plus radiotherapy (IMRT, 46 Gy) and Oral capecitabine (1.5g*2 Bid) | Right radical mastectomy with sentinel lymph node biopsy in ipsilateral armpit; and post-operative chemotherapy with TC*6 cycles. | Left radical mastectomy with sentinel lymph node biopsy in ipsilateral armpit; post-operative chemotherapy with AC-T*8 cycles and oral capecitabine (1.5g*2 Bid). |
| Follow-up | Loss to follow-up | Alive without relapse for 21 months. | Alive without relapse for 12 months. |

Note:
BMCA, Breast mucinous cystadenocarcinoma; IDC, Invasive Ductal Carcinoma; NOS, Not Otherwise Specified; G, Grade; ER, Estrogen Receptor; PR, Progesterone Receptor; EGFR, Epidermal Growth Factor Receptor; HER-2, Human Epidermal Growth Factor Receptor 2; Ki-67 - Antigen Kiel 67; pTNM, Pathological Tumor-Node-Metastasis; ypTNM, Pathological Tumor-Node-Metastasis after Neoadjuvant Therapy; PV, Pathogenic Variant; NGS, Next-Generation Sequencing; NX, Navelbine (Vinorelbine) plus Xeloda (Capecitabine); NACT, Neoadjuvant Chemotherapy; TAC, Docetaxel, Doxorubicin (Adriamycin), and Cyclophosphamide; IMRT, Intensity-Modulated Radiation Therapy; Bid Bis in Die; TC, Docetaxel (Taxotere) and Cyclophosphamide; AC-T, Adriamycin (Doxorubicin) and Cyclophosphamide followed by Taxane.

genomic DNA (gDNA) was extracted from tissues using a standard DNA extraction kit (FirePureTM FFPE gDNA Extraction Kit #FG0323; FireGen, Haidian, China), as previously described (*Yang et al., 2023*). DNA concentration was determined using Qubit dsDNA HS Assay Kit (Thermo Fisher Scientific, Waltham, MA, USA), with a minimum concentration requirement of 20 ng/μL. DNA integrity was assessed by agarose gel electrophoresis to ensure fragment size distribution above 500 bp. Exome capture libraries were prepared using the Hieff NGS Ultima Pro DNA Library Prep Kit for Illumina and SSELXT CRE V4 capture kit, following the manufacturer's instructions. Whole-exome sequencing was performed on the DNBSEQ-T7 platform (MGI Tech, Shenzhen, China), generating paired-end 150 bp reads at >100× mean coverage.

Raw sequencing data were filtered using fastp to remove adapters and low-quality reads. Only data with Q30 (Quality Score 30) ≥85% were retained for analysis. Clean reads were aligned to the GRCh38 (Genome Reference Consortium Human Build 38) human

reference genome using sentieon bwa mem (default parameters), and polymerase chain reaction (PCR)/optical duplicates were marked with the Genome Analysis Toolkit (GATK) MarkDuplicates (parameters: –OPTICAL_DUPLICATE_PIXEL_DISTANCE 2500, –ASSUME_SORT_ORDER "coordinate", –CLEAR_DT false, –CREATE_MD5_FILE true). Variants (Single Nucleotide Variants (SNVs) and indels) were identified using GATK with parameter-T to set threshold values.

MuSiC2 software was used to analyze significantly mutated genes, and at least two genes with a false discovery rate (FDR) less than 0.05 were identified as having significant mutations (Dees et al., 2012).

Downstream analyses included the identification of copy number variations (CNVs), gene rearrangements, and other genomic alterations, following previous studies (Lorente-Bermúdez et al., 2024; Wu et al., 2024).

## Immunohistochemistry

Immunohistochemical staining was performed using a Ventana Benchmark System (BenchMark ULTRA; Ventana Medical Systems, Tuscon, AZ, USA) with an UltraView Universal DAB (diaminobenzidine) Detection Kit for visualization. The primary antibody (BMP2, polyclonal, 1:200 dilution) was obtained from Boster Biological Technology Co., Ltd. (Wuhan, China).

## Bioinformatics for significant mutational gene in breast cancer patients

The significantly mutated gene BMP2 was analyzed using the cBioportal website (https://www.cbioportal.org/), including mutation mapper, pathological characteristics, correlation analysis, methylation information, survival analysis, mRNA (messenger ribonucleic acid) expression, mutation count, immunocyte infiltration level, Venn diagram, and enrichment analysis.

## Statistical analysis

All data were statistically analyzed using SPSS (Statistical Package for the Social Sciences) 17.0 software for non-parametric tests. The Wilcoxon test was used to compare two groups and the Kruskal test was used for the global comparison of multiple groups. Statistical significance was set at $P < 0.05$.

## RESULTS

### Clinical characteristics and pathological findings of patients with MCA

The clinicopathological characteristics of the three cases of breast MCA are presented in Table 1. The median age at diagnosis was 55 years (range 53–66 years). All three patients were post-menopausal. In case 1, computed tomography (CT) and magnetic resonance imaging (MRI) revealed a large mass (11.6 cm) with its greatest diameter in the outer region of the right breast, with unevenly increased intensity, which had invaded the chest wall and skin. A mass in the left breast was detected simultaneously after admission to our hospital (Supplemental S1A–S1C), whereas pelvic MRI indicated no lesions

(Supplemental S1D). However, in cases 2 and 3, a solid cystic mass with unevenly increased intensity was detected by MRI in the upper outer quadrant (UOQ) of the right breast (3.5 cm in its greatest diameter) and upper inner quadrant (UIQ) of the left breast (3.0 cm in its greatest diameter), respectively, without a contralateral tumor identified simultaneously (Supplemental S2A, S2B). Enlarged lymph nodes >1 cm were found in the right auxiliary region of case 1, which were absent in cases 2 and 3 at the time of initial diagnosis. In case 1, the patient received neoadjuvant chemotherapy with seven cycles of docetaxel, doxorubicin (adriamycin), and cyclophosphamide (TAC) and three cycles of Navelbine with cisplatin (NP) before modified radical mastectomy of the left and right breasts, respectively. Pathological examination confirmed the diagnosis of bilateral breast cancers: infiltrating ductal carcinoma, not otherwise specified (IDC, NOS), grade 3 (Nottingham Grade), G1 (Miller-payne Grade) in the left breast and breast MCA, grade 2 (Nottingham Grade), and G1 (Miller-payne Grade) in the right breast with metastasis to the skin of the right chest wall. In cases 2 and 3, breast MCA was detected in the unilateral breast (right and left breast, respectively), and the patients underwent radical mastectomy with sentinel lymph node biopsy in the ipsilateral armpit plus post-operative chemotherapy; both were in pT2N0M0 stage IIA. After 21 and 12 months of follow-up respectively, the two patients showed no evidence of relapse or metastasis. However, in case 1, the patient experienced a dismal clinical course and received neoadjuvant chemotherapy (NACT) with seven cycles of TAC and three cycles of NP prior to surgery because of the extensive tumor burden of bilateral breast cancers. This was followed by an extended radical mastectomy for both the left and right breast cancers, sequentially. Postoperative treatment included seven cycles of navelbine (vinorelbine) plus xeloda (capecitabine) (NX) chemotherapy, radiotherapy (intensity-modulated radiation therapy (IMRT), 46 Gy), and oral capecitabine (1.5g * 2 Bid) as maintenance therapy. Ultimately, the patient was diagnosed with ypT4N2M1 and rT4N2M1 stage IV breast cancers on both sides in case 1. Within 2 years of surgery and systemic treatment, she developed multiple metastases in the right chest wall and was subsequently lost to follow-up.

On gross examination, median maximal diameter of breast cancer in the three cases was 3.75 cm (range = 3.0–15.0 cm), mostly with a well-circumscribed solid and cystic mass. Microscopically, different proportions of MCA components were observed: case 3 exhibited pure breast MCA, grade 3 (Nottingham Grade); case 2 featured a mix with IDC, NOS, grade 2 (Nottingham Grade); and case 1 presented synchronous bilateral breast cancers: IDC, NOS, grade 3 (Nottingham Grade), G1 (Miller-payne Grade) in the left breast, and breast MCA, grade 2 (Nottingham Grade), G1 (Miller-payne Grade) in the right breast with metastasis to the skin of the right chest wall). These MCA components displayed characteristic histological features, such as a single layer or pseudostratified tall columnar epithelial cells of low to intermediate nuclear grade and complex branched mucinous papillary structures, closely resembling those found in their ovarian or pancreatic counterparts (Figs. 1, 2). However, tumor tumor-infiltrating lymphocytes (TILs) in the stroma between the areas of carcinoma cells, including all mononuclear cells (lymphocytes and plasma cells), were less than 5% in all cases. Macro-metastasis in eight

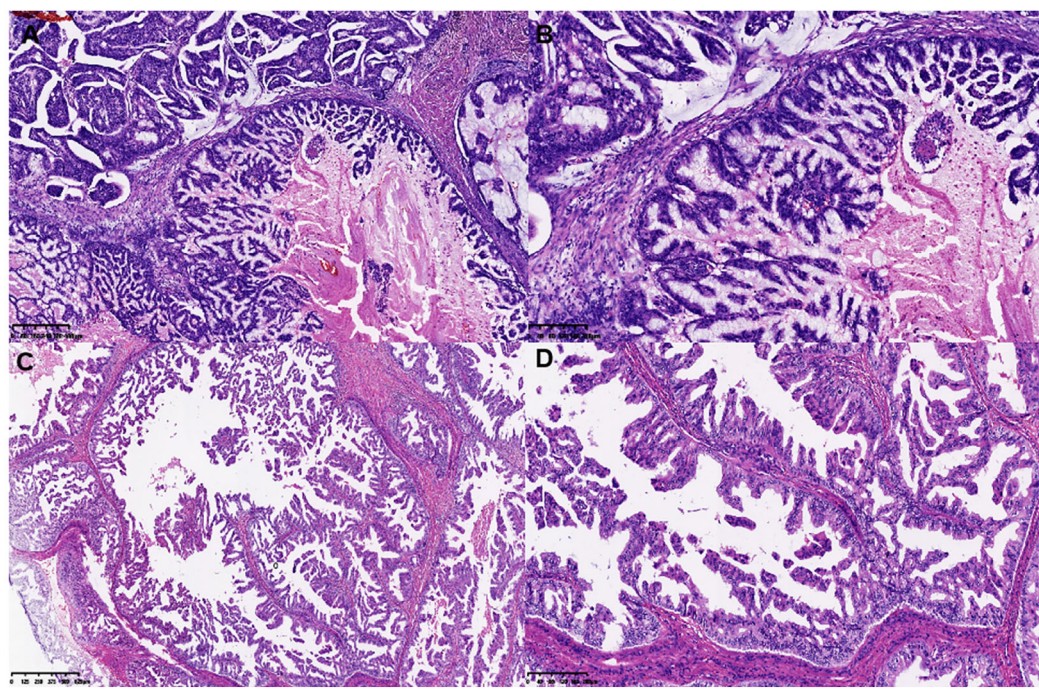

**Figure 1 Histological features of MCA.** (A and B) Histological features of single layer or pseudos-tratified tall columnar epithelial cells, and (C and D) complex branched mucinous papillary structures at low and high power microscopically in MCA.

lymph nodes was identified in 26 right auxiliary lymph nodes, and no metastasis was found in the left in 26 right auxiliary lymph nodes (metastasis-positive lymph node number *vs.* total lymph node number: 0/33) of case 1, while no metastasis was found in the lymph nodes of cases 2 and 3 (metastasis-positive lymph node number *vs.* total lymph node number: 0/7, 0/12 respectively). In cases 2 and 3, high-grade ductal carcinoma *in situ* (DCIS) components were observed adjacent to the invasive tumor, corroborating the diagnosis of primary breast MCA. However, because the bilateral breast tumors in case 1 lacked precursor components, metastasis exclusion was crucial for accurate diagnosis. Immunostaining showed that the epithelial cells in the breast MCA were positive for cytokeratin 5/6 (CK5/6) (focal) (Fig. 3A), cytokeratin 7 (CK7) (Fig. 3B), and epidermal growth factor receptor (EGFR) (Fig. 3C), but negative for ER, PR, HER-2, CK20 (cytokeratin 20) (Fig. 3D), Caudal Type Homeobox 2 (CDX2) (Fig. 3E), and PAX8 (Paired Box Gene 8) (Fig. 3F), which was in accordance with primary breast MCA. In addition, a positron emission tomography/computed tomography (PET/CT) scan performed before surgery in case 1 also indicated a primary bilateral breast mass with extensive auxiliary lymph node metastasis and no signs of visceral organ lesions. Cases 2 and 3 also demonstrated a triple-negative (hormone receptor of ER, PR negative, HER2-negative) immunophenotype, and a high Ki-67 index was observed in three cases (40–60%, with a median value of 40%). The metastatic origin was also ruled out by additional imaging examinations in cases 2 and 3.

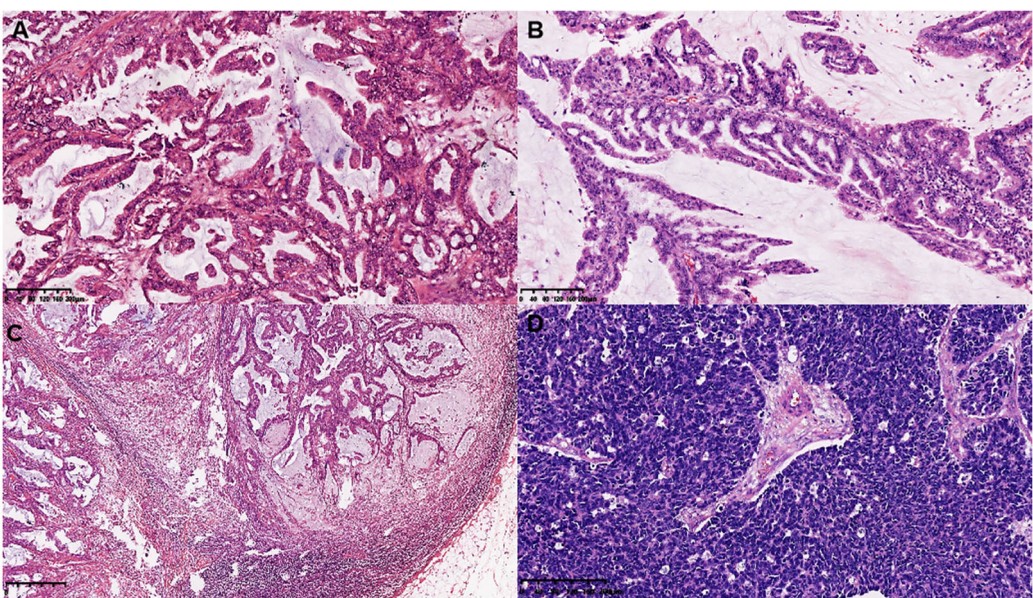

**Figure 2 Histomorphology of the MCA, macro-metastasis in lymph nodes, and histomorphology of contralateral triple negative breast cancer in case 1.** (A and B) Histomorphology of the MCA cells in case 2 and case 3. (C) Macro-metastasis in lymph nodes, and (D) histomorphology of contralateral triple negative breast cancer of case 1.

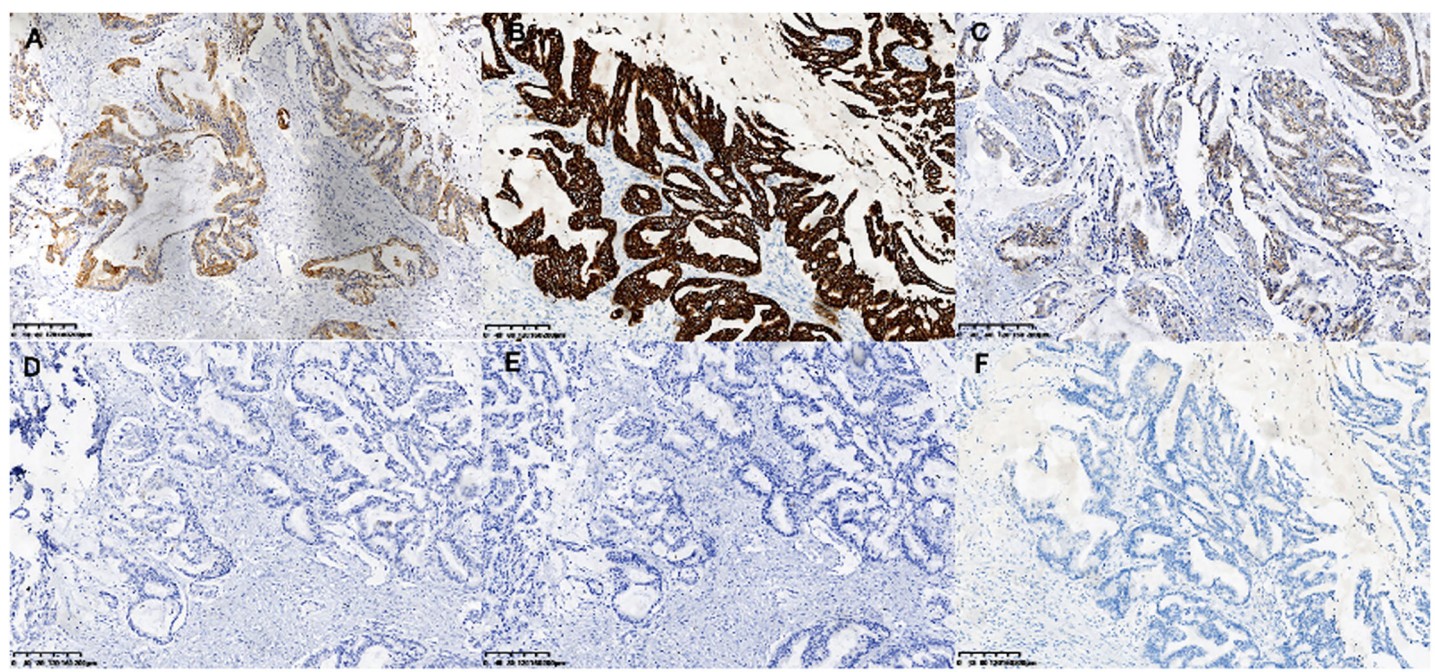

**Figure 3 Immunohistochemical staining of MCA.** Immunohistochemical staining (×200) shown the tumor cells were positive for CK5/6 (A), CK7 (B) and EGFR (C), and negative for CK20 (D), CDX2 (E) and PAX-8 (F).

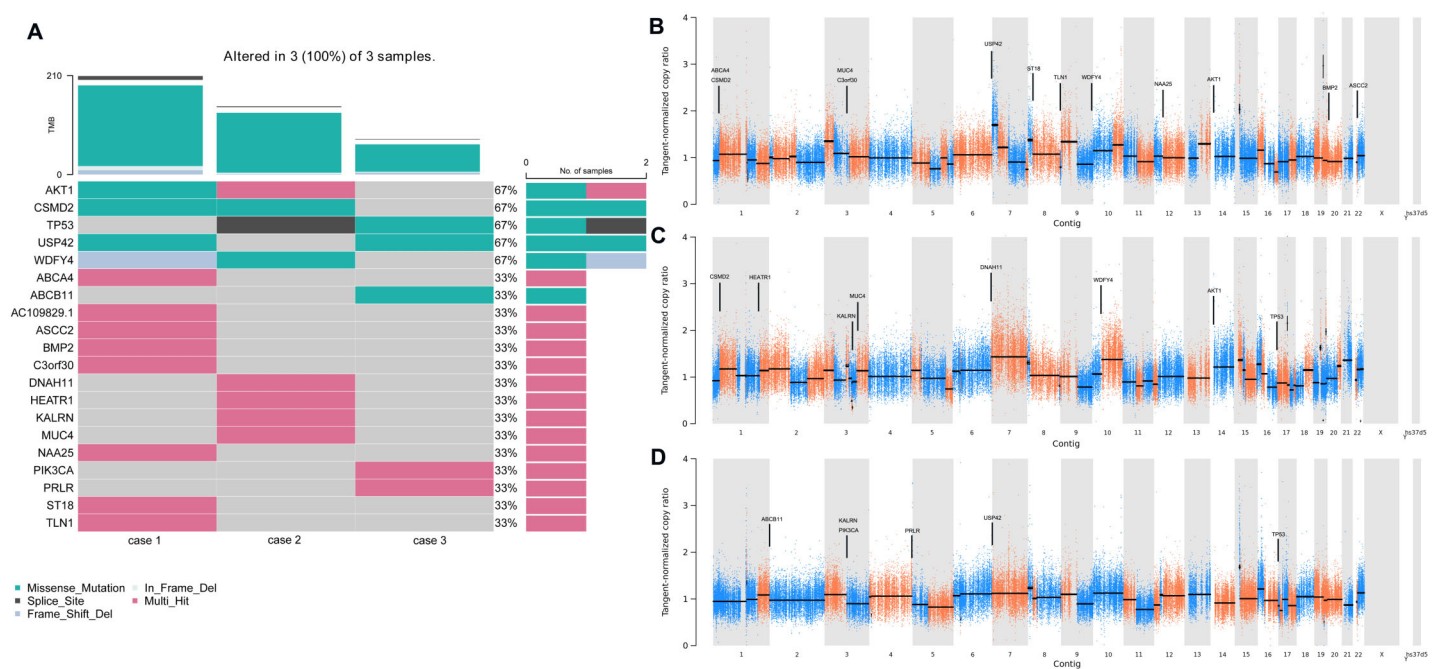

**Figure 4 Gene alteration and significant mutant genes in MCA patients by WES.** (A) Heatmaps depicted the somatic genetic alterations identified by WES. Cases were shown in columns and genes in rows, besides, alterations were color-coded according to the legend. (B–D) Genome plots depicted tangent-nomalized copy ratios (y-axis) plotted according to their genomic positions (x-axis) of case 1–3.

## Gene alteration and significant mutant genes in MCA patients by WES

To determine the repertoire of genetic alterations in MCA, we subjected these three cases to WES. However, WES analysis revealed a scarcity of CNVs and somatic mutations in the corresponding chromosomes (Figs. 4A, 4B); the most commonly mutated genes were AKT Serine/Threonine Kinase 1 (AKT1), CUB and Sushi Multiple Domains 2 (CSMD2) TP53, Ubiquitin Specific Peptidase 42 (USP42), WD Repeat and FYVE Domain Containing 4 (WDFY4) (each altered in two out of the three cases). The tumor mutation burden in the three cases were all below 10 mutations/megabase (Mb), which was 6.21, 4.35, 1.92 mutations/Mb respectively. All three cases were in a microsatellite-stable status. Notably, in case 1, BMP2 was identified by MuSiC2 (false discovery rate (FDR) = 0.031) as a significant mutant gene (SMG) from the WES data of three patients, although its significance remains unclear. To explore whether BMP2 could serve as a substrate for the development of breast MCA, we compared the expression level of BMP2 in breast MCA with that in triple-negative breast cancer (TNBC) and the histology of IDC by immunohistochemistry. As shown in Fig. 5, BMP2 staining was positive across all MCA cases in our study (Figs. 5A, 5B); however, TNBC patients with IDC histology exhibited a faint staining pattern (Figs. 5C, 5D).

## BMP2 is associated with tumor progression in breast cancer

To further explore the function of BMP2 in breast cancer, we used the cBioPortal online tool to analyze functional genomics. In The Cancer Genome Atlas (TCGA) and the

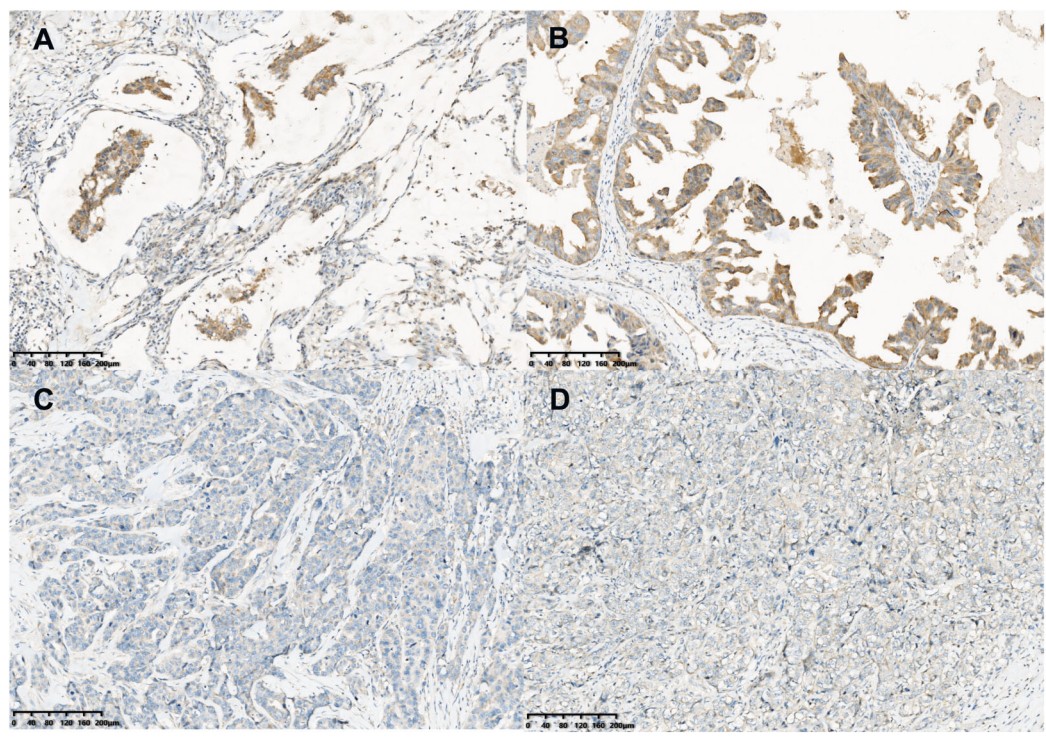

**Figure 5 BMP2 staining in MCA.** Immunohistochemical staining (×200) showed the tumor cells were moderately positive for BMP2 (A and B) in breast MCA, while faint BMP2 staining was observed in the patients with TNBC of IDC histology (C and D).

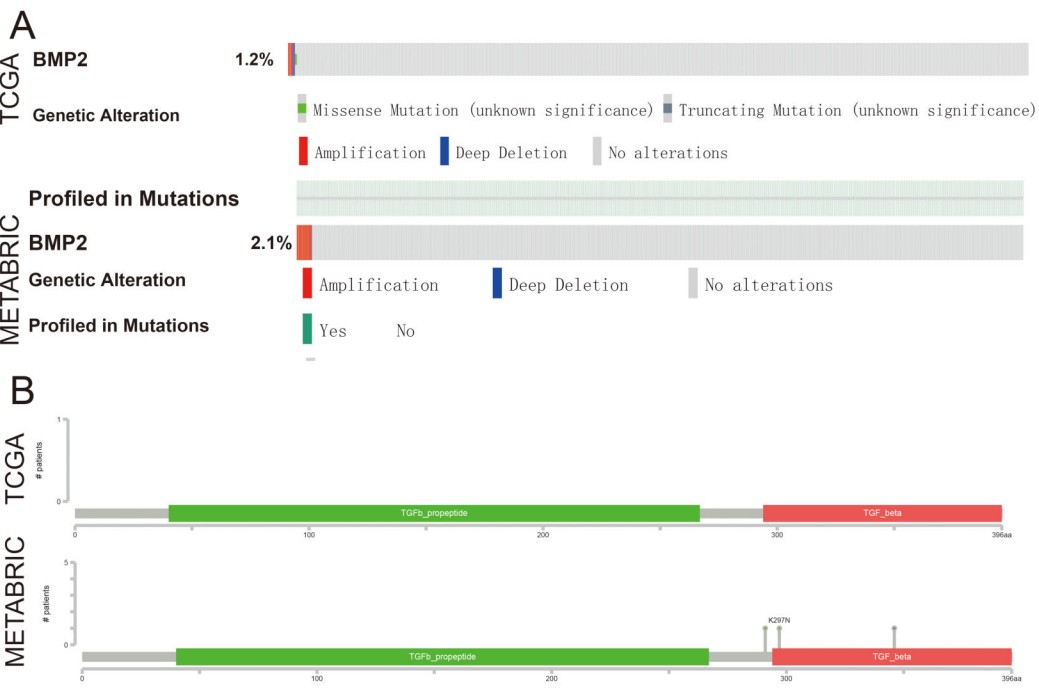

**Figure 6 The genetic alteration and mutation mapper of BMP2.** (A) The ratio of BMP2 genetic alteration in breast cancer patients of TCGA and METABRIC database. (B) Mutation mapper of mutation site on BMP2.

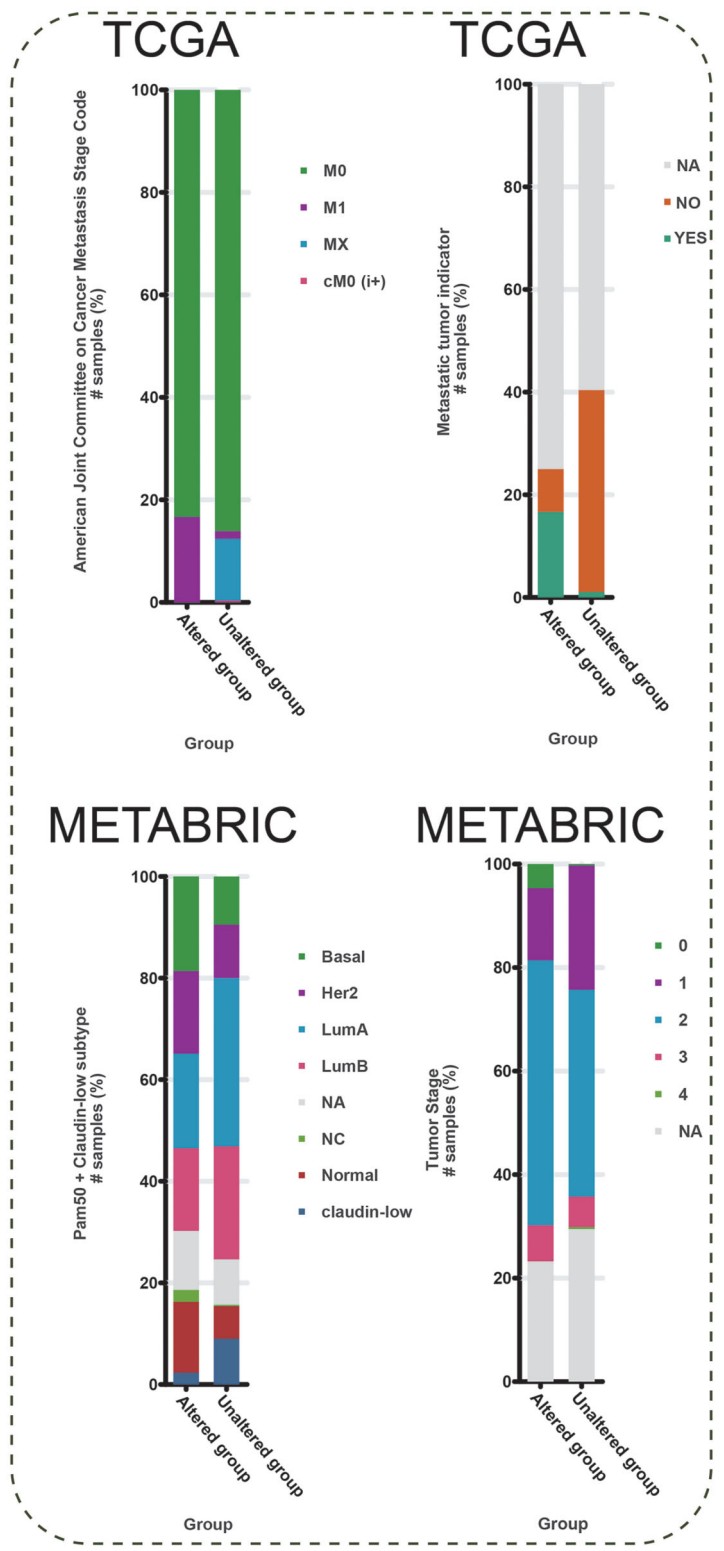

**Figure 7 Metastasis stage code, metastatic tumor indicator, Pam50 subtype, and stage of BMP2.** Metastasis stage code, metastatic tumor indicator, Pam50 subtype, and stage, in altered or unaltered group of BMP2 in breast cancer patients of METABRIC database.

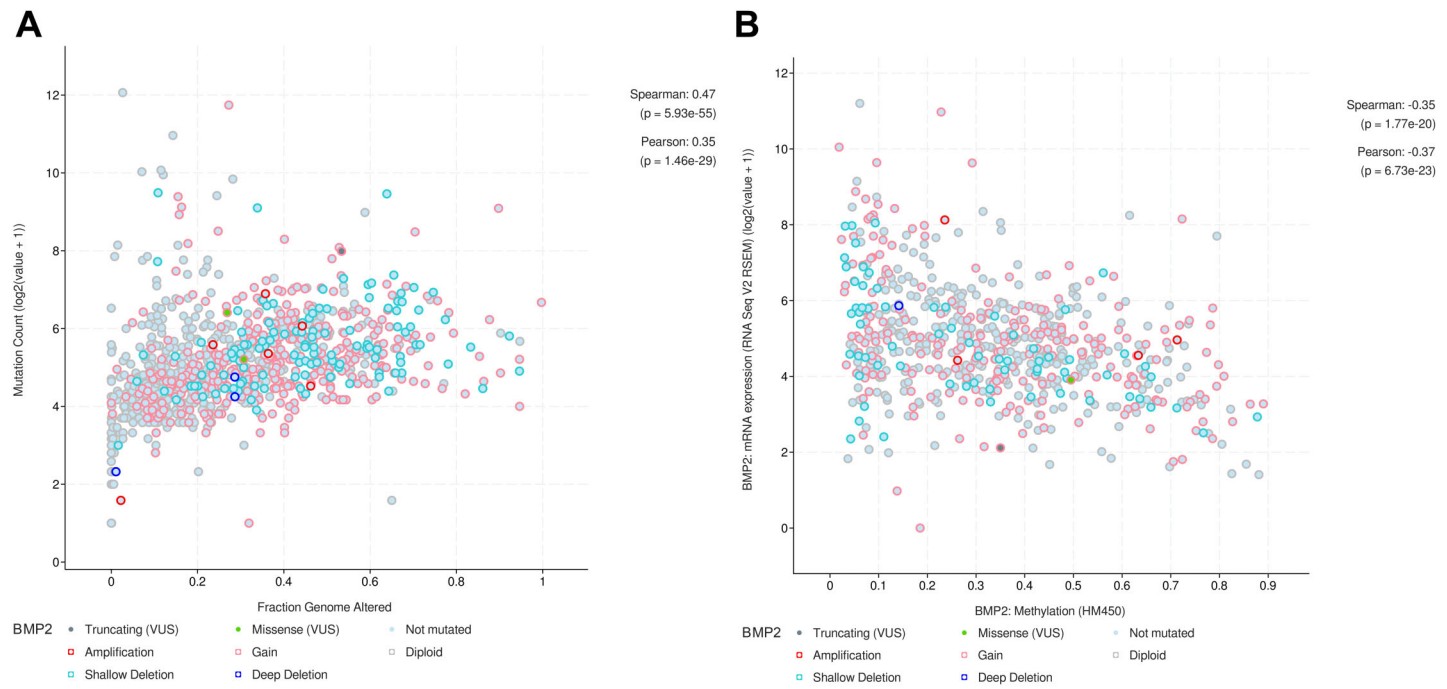

**Figure 8 Correlation between fraction genome altered and mutation or methylation status and the mRNA expression of BMP2.** (A) Positive correlation between fraction genome altered and mutation count of BMP2. (B) Negative correlation between methylation status and the mRNA expression of BMP2.

Molecular Taxonomy of Breast Cancer International Consortium (METABRIC) databases, the frequencies of BMP2 mutations were 1.2% and 2.1%, respectively, with amplification and deep deletion being the main types of gene alterations (Fig. 6A). The location of BMP2 alteration is K297N on transforming growth factor-beta (TGF-b) propetide in METABRIC database (Fig. 6B). As depicted in Fig. 7A, BMP2 alteration was associated with tumor metastasis status and metastatic tumor indicators in TCGA database and correlated with Prediction Analysis of Microarray 50 (PAM50)-based molecular subtype and tumor stage in the METABRIC database. A positive correlation between Fraction genome altered and mutation count of BMP2 was identified (Fig. 8A). Additionally, methylation status negatively correlated with the mRNA expression of BMP2 (Fig. 8B).

Next, the frequency scatter of enrichments in the alteration of BMP2 in breast cancer patients was higher than that of the unaltered ones in the TCGA and METABRIC databases (Figs. 9A, 9B). One hundred and forty-seven related genes of BMP2 were identified in both the TCGA and METABRIC databases (Fig. 10A). The network of 147 related genes is depicted in Fig. 11A. The enrichment analysis of BMP2 and its 147 related genes is shown in Fig. 12A, encompassing the Reactome, Kyoto Encyclopedia of Genes and Genomes, cellular components, molecular functions, and biological processes.

Finally, we examined the effect of BMP2 expression on the survival rate of patients with breast cancer. Patients with genetic alterations in BMP2 showed reduced survival, as

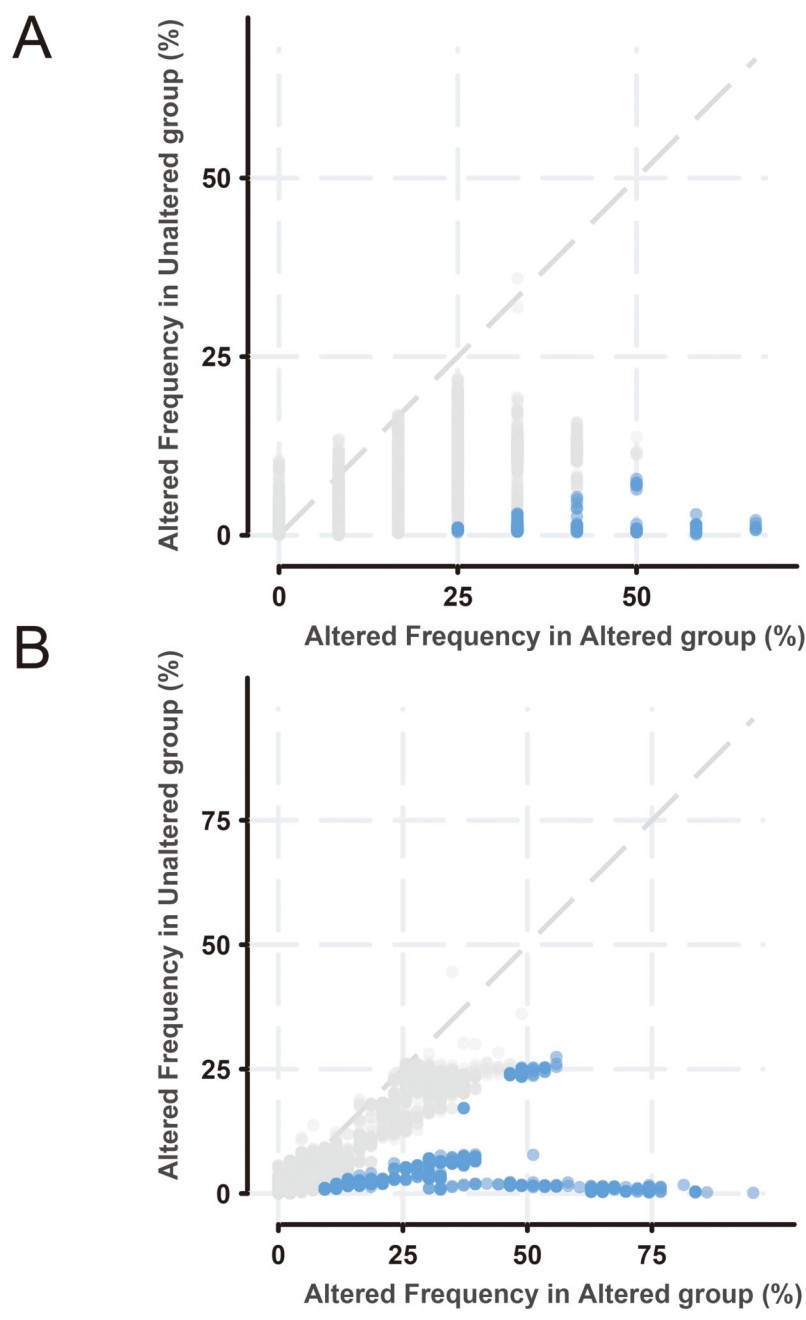

**Figure 9 Frequency scatter of enrichments of BMP2.** (A and B) Frequency scatter of enrichments in altered or unaltered group of BMP2 in breast cancer patients of TCGA and METABRIC database.

demonstrated by overall survival analysis in the TCGA database (Fig. 13A) and relapse-free status analysis in the METABRIC database (Fig. 13C). Although the overall survival of the BMP2-altered group was lower than that of the BMP2-unaltered group in the METABRIC database, this trend was not statistically significant (Fig. 13B). Analysis of
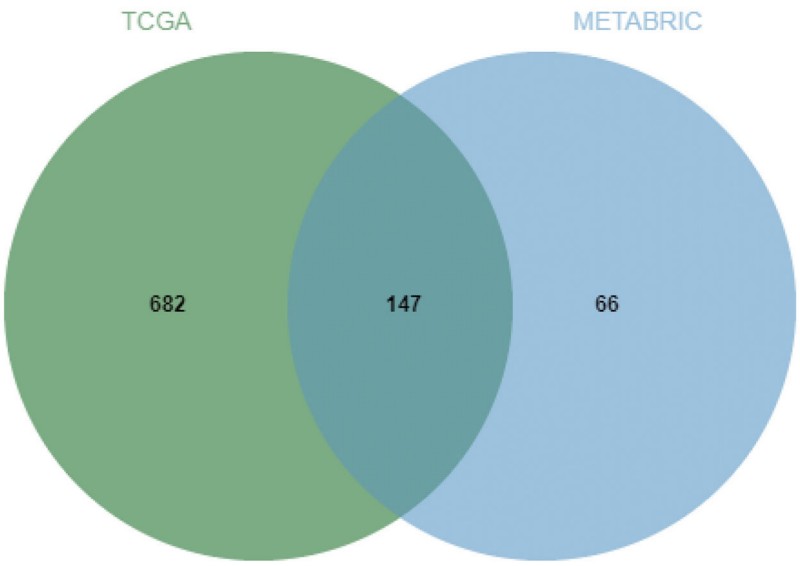

**Figure 10  Venn diagram of related genes of BMP2.** 

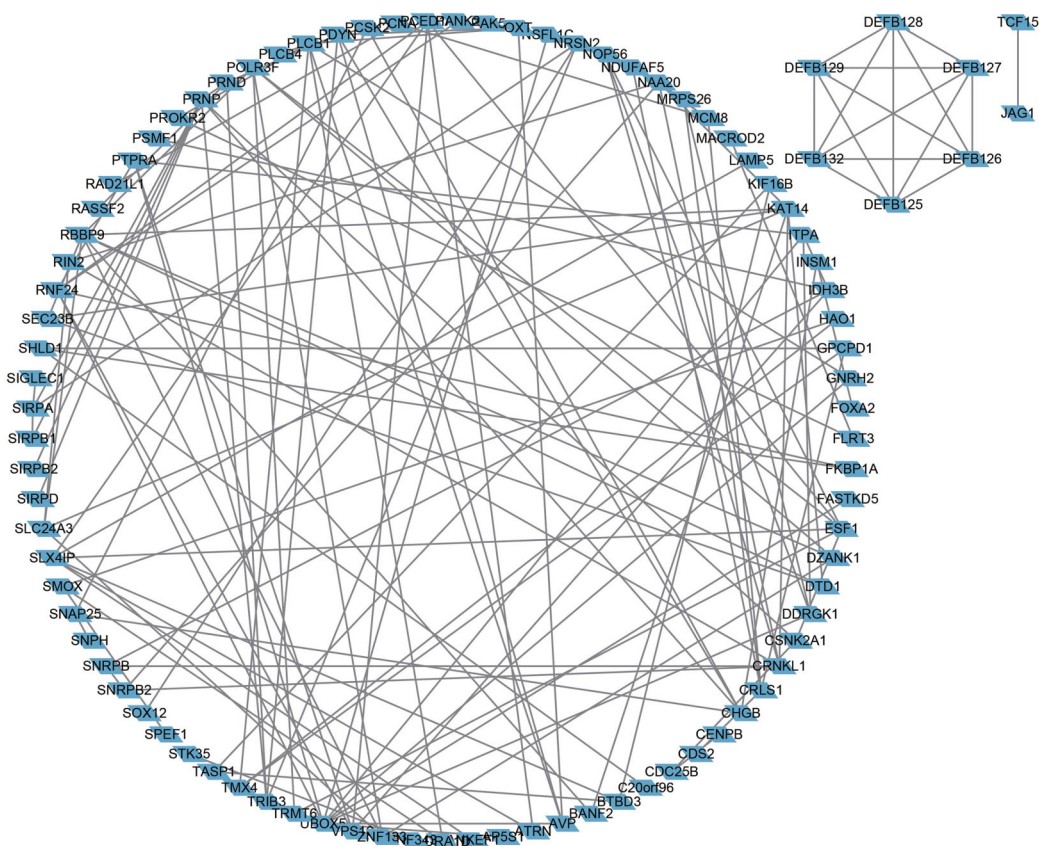

**Figure 11  Network of 147 related genes of BMP2.** Network of 147 related genes of BMP2 in breast cancer patients of TCGA and METABRIC database. 

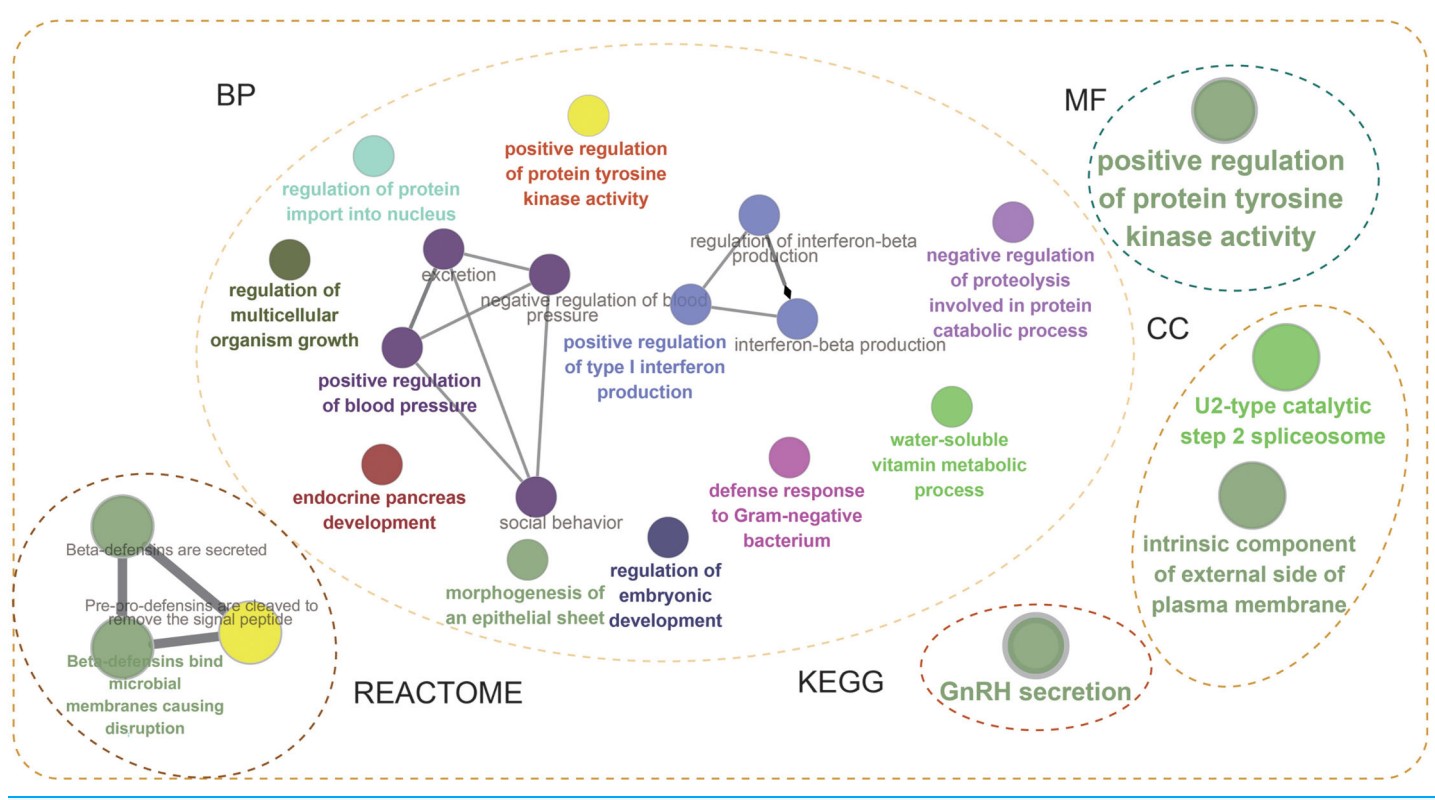

**Figure 12 Enrichment analysis of BMP2.** Enrichment analysis of BMP2 and its 147 related genes.

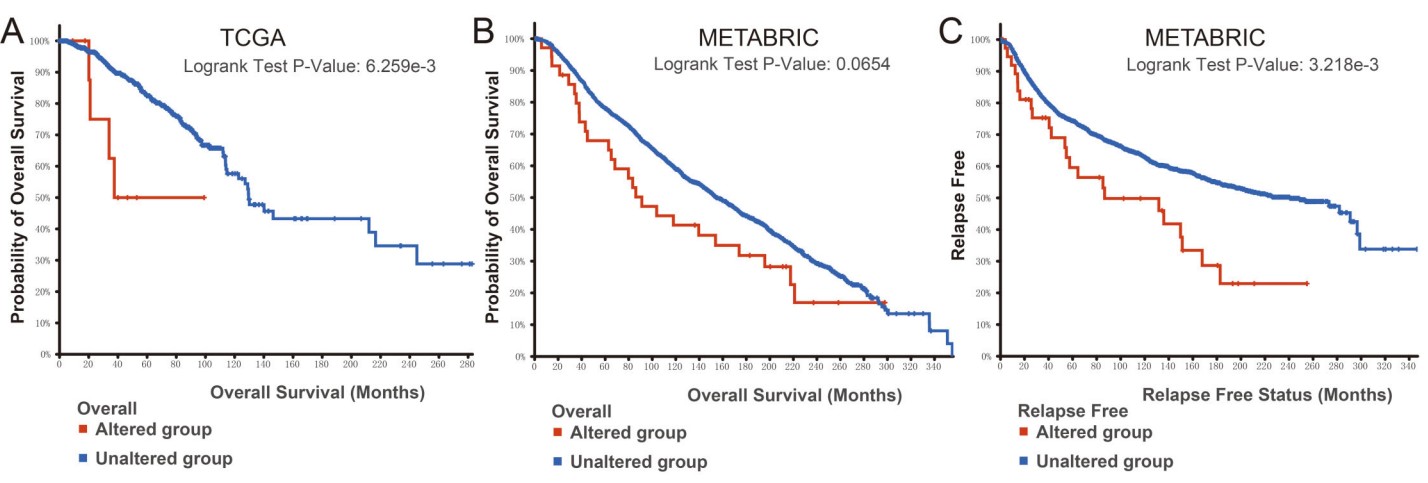

**Figure 13 Survival analysis of patients of BMP2.** (A–C) Survival analysis of patients in altered or unaltered group of BMP2 of TCGA and METABRIC database.

the correlation between mRNA expression and CNV from the Genomic Identification of Significant Targets in Cancer revealed that the mRNA expression of BMP2 was associated with shallow deletion, diploidy, and gain in the TCGA database (Fig. 14A). Similarly, the mutation count and CNV from DNAcopy demonstrated that the BMP2 mutation was

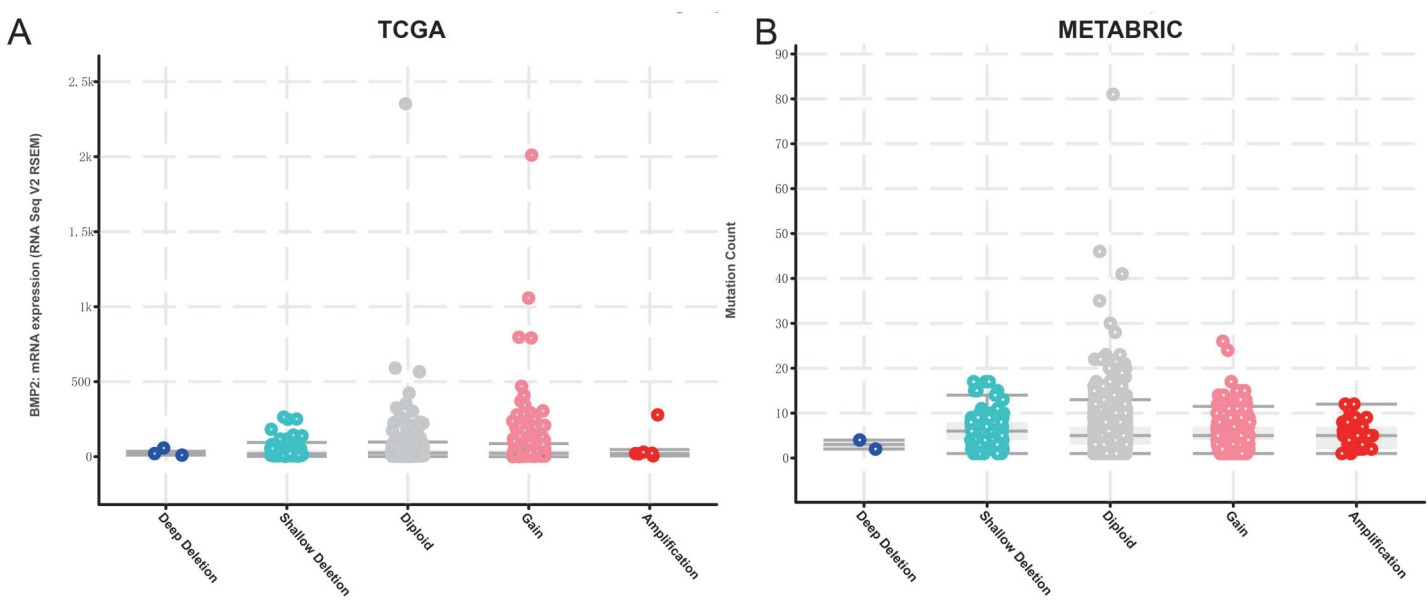

**Figure 14 mRNA expression and mutation count of BMP2.** (A) BMP2 mRNA expression and mutation count in different categories of genetic alteration of BMP2.

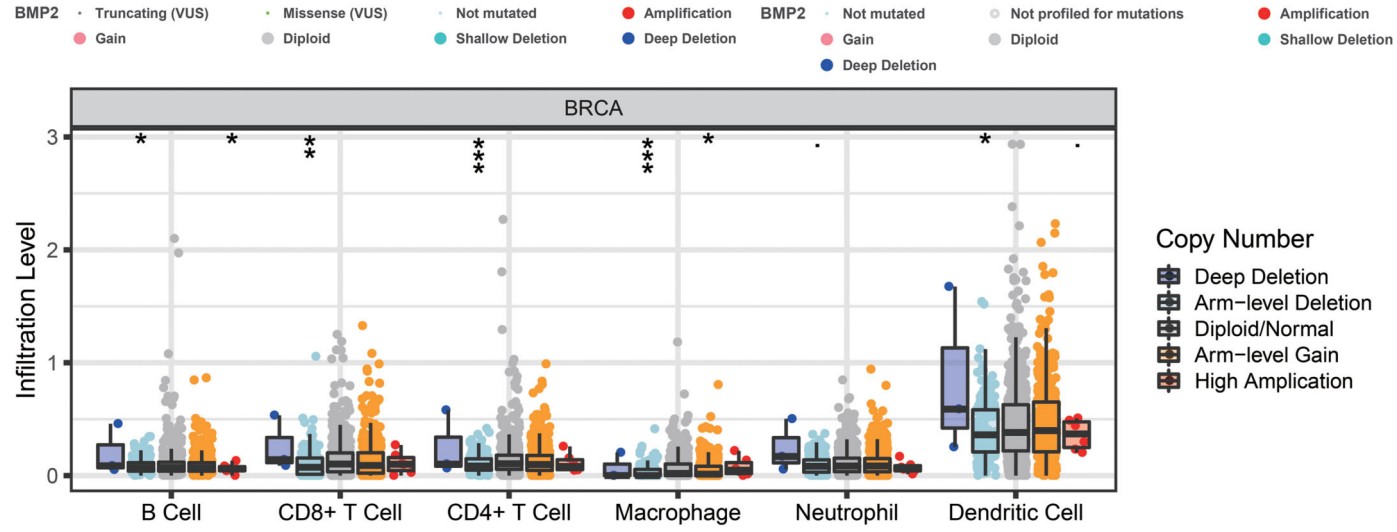

**Figure 15 The infiltration level of various immunocytes of BMP2.** The infiltration level of various immunocytes in different categories of genetic alteration of BMP2 of breast cancer patients. $^*p < 0.05$, $^{**}p < 0.01$, $^{***}p < 0.001$.

associated with shallow deletion, diploidy, and gain in the METABRIC database (Fig. 14B). It should be pointed out that genetic alterations of BMP2 were also linked to immune cells, including B cells, CD8 (Cluster of Differentiation 8)+ T cells, CD4 (Cluster of Differentiation 4)+ T cells, macrophages and dendritic cells, particularly in cases of arm-level deletion of BMP2 alterations (Fig. 15A).

## DISCUSSION

Mucinous cystadenocarcinomas (MCAs) predominantly arise in the ovary, pancreas, and gastrointestinal tract, while primary breast MCAs are exceedingly rare. The presence of DCIS components in the adjacent mammary tissue suggests a potential histological relationship and provides important evidence supporting a mammary origin for these tumors. Owing to their rarity, a comprehensive workup that includes clinical history, imaging, and extensive immunohistochemical profiling is essential to exclude metastatic MCAs and other mucinous tumors to reach an accurate diagnosis (*Nissan et al., 2025*). Within our cohort, Case 1 was a recurrent MCA of the right breast with synchronous contralateral synchronous IDC, which showed distinct clinical behaviors contrasting sharply with the indolent course typically reported in mammary MCA (*Joneja & Palazzo, 2023*; *Lei, Shi & Chen, 2023*). This patient exhibited an unusually heavy tumor burden with chest wall metastasis and a dismal prognosis after surgery and systemic therapy for nearly 2 years, highlighting the potential complexity and heterogeneity of breast MCAs. The high Ki67 index (60%) in case 1 may be linked to a poor response to neoadjuvant and post-operative chemotherapy, although the precise molecular mechanisms underlying this aggressive phenotype remain to be elucidated.

While initially established as a critical signaling molecule involved in numerous physiological and pathological processes including bone formation, embryonic development, and tissue homeostasis (*James et al., 2016*; *Lavery et al., 2008*; *Seeherman et al., 2019*; *Tanaka et al., 2014*), Bone morphogenetic protein 2 (BMP2) has emerged as a molecule of significant interest in oncological research. It has been reported to be implicated in the pathogenesis of a wide range of malignancies, including colorectal cancer, non-small cell lung cancer, renal cell carcinoma, pancreatic cancer and breast cancer, where its role varies between tumor-promoting and tumor-suppressing functions (*Bach, Park & Lee, 2018*; *Huang et al., 2021*; *Shimizu et al., 2023*). The role of BMP-2 in the tumorigenesis and metastasis of breast cancer remains controversial, with most studies indicating a pro-tumorigenic function for this molecule, which appears to be influenced by the specific breast cancer subtype and the tumor microenvironment (*Buijs et al., 2012*; *Chapellier et al., 2015*; *Clement et al., 2005*; *Huang et al., 2017*; *Ihle et al., 2024*; *Liu et al., 2023*; *Pouliot & Labrie, 2002*; *Ye, Bokobza & Jiang, 2009*). In this study, as a SMG based on WES data analysis, two novel in-frame deletions of BMP2 (p.Val381del and p.Leu382_Tyr385del) were identified in the tumor sample of case 1, which have not been reported previously. However, whether alterations in the amino acid level of BMP2 influence the tertiary structure of the protein, secretion, and clearance in the circulation remains unknown. While our bioinformatics analysis demonstrated that BMP2 transcriptional alterations correlated significantly with tumor metastasis, immune cell infiltration, and poor survival in patients with breast cancer, there still exists a substantial gap between genetic findings and experimental validation. Our preliminary immunohistochemistry study showed moderate BMP2 expression in breast cystadenocarcinoma, whereas IDC with the TNBC phenotype exhibited only modest expression, which requires further validation with larger cohorts. A comprehensive

understanding of the mechanistic role of BMP2 in breast MCA could enhance our understanding of tumor biology and facilitate the development of novel therapeutic strategies, with a potential to improve clinical outcomes in this rare tumor type.

The etiology and pathogenesis of breast MCA remain largely unknown, due to its rarity and the limited number of studies investigating genetic changes in breast MCA. In our study cohort, as illustrated in Fig. 4, WES analysis revealed that AKT1, CSMD2, TP53, USP42, and WDFY4 were the most frequently mutated alleles, each detected in two out of three cases. Other top 20 mutated genes, such as ATP Binding Cassette Subfamily A Member 4 (ABCA4) and ATP Binding Cassette Subfamily A Member 11 (ABCB11), appeared less frequently, being identified in only one case each. Among the identified candidates, TP53, AKT1, Mucin 4 (MUC4) and PIK3CA are known driver genes, while the functional significance of the remaining 16 identified SNVs warrants further exploration. As breast mucinous adenocarcinoma and cystadenocarcinoma are two distinct entities with unique clinical and pathological features, the precursor lesion of breast cystadenocarcinoma remains undefined (Joneja & Palazzo, 2023). In contrast, the counterparts in the ovary, pancreas, and appendix are believed to originate from borderline mucinous tumors (Cheasley et al., 2019; Liao et al., 2020; Zamboni et al., 1999). In ovarian mucinous tumors, cyclin dependent kinase inhibitor 2A (CDKN2A) and KRAS mutations represent the most common genetic alterations in both ovarian mucinous adenocarcinoma and their precursor lesions. In addition, TP53 mutations and HER2 amplifications may promote the malignant progression of ovarian borderline mucinous tumors (Cheasley et al., 2019; Hada et al., 2023). Similarly, KRAS mutations are the most frequent genetic alterations in mucinous tumors of the pancreas and appendix, while mutations in APC (Adenomatous Polyposis Coli), TP53, and SMAD Family Member 4 (SMAD4) have been associated with high-grade or invasive disease (Liao et al., 2020; Zamboni et al., 1999).

Our study characterized three cases of primary breast MCA and provided detailed clinicopathological and genetic characteristics. Despite the pathogenesis and etiology of this uncommon entity of breast cancer remain incompletely understood, the BMP2 alterations identified in our study represents a potentially significant finding for further investigation. The primary limitations of this investigation include the inherently small cohort of patients due to the low incidence and absence of RNA- and protein-level data for further integrated analysis. Although extensive efforts were made in experimental design, data processing, and quality control to minimize the impact of confounding factors, it is still difficult to fully eliminate the influence of unmeasured environmental variables, which may introduce bias into the results and lead to an overestimation of the genetic contribution. Future multi-institutional collaborative studies with larger patient cohorts and comprehensive investigations would help enhance statistical power and the generalizability of our findings, and facilitate the development of predictive biomarkers, risk stratification, and personalized treatment, and ultimately improved outcomes for patients with this rare breast cancer subtype.

## CONCLUSIONS

In summary, our findings expanded the current understanding of breast MCA by elucidating its clinical and molecular characteristics, underscoring the significance of tumor stage as a critical determinant of patient outcomes. Our data identified BMP2 as a potential driver of tumor progression, highlighting the need for additional studies in larger cohorts. These findings may pave the way for improved prognostic assessment and novel therapeutic strategies targeting BMP2 in breast MCA.

### Funding

This work was funded by the Senior Medical Talents Program of Chongqing for Young and Middle-aged (2020-219). The funders had no role in study design, data collection and analysis, decision to publish, or preparation of the manuscript.

### Grant Disclosures

The following grant information was disclosed by the authors:
Senior Medical Talents Program of Chongqing for Young and Middle-aged (2020-219).

### Competing Interests

The authors declare that they have no competing interests.

### Author Contributions

- Zhenyu Li conceived and designed the experiments, analyzed the data, prepared figures and/or tables, and approved the final draft.
- Yi Gong analyzed the data, authored or reviewed drafts of the article, and approved the final draft.
- Guangxin Li performed the experiments, authored or reviewed drafts of the article, and approved the final draft.
- Qingming Jiang conceived and designed the experiments, performed the experiments, authored or reviewed drafts of the article, and approved the final draft.
- Juanhui Dong performed the experiments, authored or reviewed drafts of the article, and approved the final draft.
- Rui Chen conceived and designed the experiments, analyzed the data, prepared figures and/or tables, and approved the final draft.

### Human Ethics

The following information was supplied relating to ethical approvals (*i.e.*, approving body and any reference numbers):

Informed consent was obtained, and protocols were approved by the Ethics approval of Chongqing University Cancer Hospital (No. CZLS2023062-A).

## Data Availability

Data is available at GenBank: PRJNA1202584.

## Supplemental Information

Supplemental information for this article can be found online at http://dx.doi.org/10.7717/peerj.19948#supplemental-information.

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
