# Peer review of "BMP2 alterations in mucinous cystadenocarcinoma of the breast: insights from whole-exome sequencing"

_PeerJ, doi:10.7717/peerj.19948_

## Round 0.1 · original submission · Major Revisions

· Academic Editor

Major Revisions

Reviewer 1 ·

Basic reporting

The current submission is a focused paper that performs necessary analysis to explain the driver mutations of Mucinous cystadenocarcinoma of the breast. Especially the alterations in the genome that would explain the tumorigenesis. The article is written in clear language to explain the clinical characteristics and pathological findings. The introduction is abnormally short, and some of the material that should have been included in the introduction was presented in other sections, such as results and discussions. The figures are relevant and of high quality. The raw data was not needed as the information is public for most figures, except Figure 4. The raw data for WES analysis might not be able to be shared due to it being protected information. The article does not have any specific hypotheses defined for the analysis of breast cancer cases.

Experimental design

It is unclear whether the research meets the scope of PeerJ. This article is primarily a case study with some additional bioinformatics analysis for the cases analyzed in the study. The whole exome sequencing and bioinformatics analysis of the cases led to some observations that might be considered as research, but this would not be classified as an original research article. The research question is not clear based on the text provided in the short introduction.

Validity of the findings

The authors do explain some details regarding the methods used for whole-exome sequencing performed on the samples. The methods and results are not explained in sufficient detail for reproducibility. The authors do not provide any details regarding the WES assay in terms of sequencing information or variant calling information for the reviewers to evaluate the validity of their findings. It would be essential to include sequencing metrics such as average read depth and mapped read percentage for each sample in a supplemental table. A more thorough reporting of WES analysis is essential to understand the negative findings claimed by the authors in the manuscript.

The results in Fig. 4A show 15 different genes that have mutations in 1 of the three cases that were analyzed using WES. While the authors claim that BMP2 was identified as a significant mutant gene, no data is showing that whether there were any other SMGs identified from the 3 patients. I would like to see the authors provide comprehensive analysis results to show that BMP2 was prioritized as the candidate with a clear rationale from the MuSiC2 results. The bioinformatics analysis in Figures 6 and 7 is dependent on the validity of the WES analysis and BMP2 as the only SMG from the analysis.

Additional comments

I commend the authors for the extensive bioinformatics analysis of the effects of BMP2 mutations on breast cancer. Unfortunately, without a clear correlation except for the expression of BMP2 in some breast cancer cells it is difficult to understand the significance of bioinformatics findings to the cases that were studied.

The authors have missed defining some acronyms in some cases, such as PV in Table 1 (assumed to be pathogenic variants). In some cases, authors have not defined the acronym the first time they use it in the manuscript IDC. I would like the authors to go over the PDF and define the acronyms the first time they are encountered in the text for all the acronyms. I would also like them to define all the acronyms in the text below Table 1.

Annotated reviews are not available for download in order to protect the identity of reviewers who chose to remain anonymous.

Reviewer 2 ·

Basic reporting

Study Design and Manuscript Contents
・The manuscript presents a well-defined research question investigating the molecular alterations in mucinous cystadenocarcinoma (MCA) of the breast through whole-exome sequencing (WES) and immunohistochemical analysis.
・The study focuses on BMP2 as a significantly mutated gene and explores its potential role in tumor progression, linking genomic alterations to clinical characteristics.
・The methodology is sound, employing standard sequencing and bioinformatics techniques. However, the sample size is small (n=3), which limits the generalizability of the findings.

Experimental design

Relevant Comments (Major Strengths & Major Weaknesses)
Strengths:
・The study is one of the few to provide molecular insights into a rare form of breast cancer.
・The integration of WES, immunohistochemistry, and bioinformatics analysis enhances the study's robustness.
・Well-documented clinical case descriptions and imaging findings support the presented data.

Validity of the findings

Weaknesses:
・The small sample size significantly limits the statistical power and generalizability.
・Functional validation of BMP2’s role in MCA is lacking, making the clinical implications of the findings uncertain.
・The discussion does not thoroughly address potential alternative explanations or limitations in the data analysis.

Additional comments

Detailed Comments
1. Title
The title is clear and informative, but could be slightly refined for conciseness:
・Suggested revision: “BMP2 Alterations in Mucinous Cystadenocarcinoma of the Breast: Insights from Whole-Exome Sequencing”
・This revision avoids redundancy while maintaining clarity.
2. Abstract
Strengths:
・Provides a concise summary of the research question, methodology, and key findings.
Suggested Improvements:
・The methods section could briefly specify the number of cases analyzed to clarify the study's scale.
・The conclusion should explicitly state the study’s key takeaways for future research.
3. Introduction
Strengths:
・Provides a good background on MCA of the breast and highlights the knowledge gap.
Suggested Improvements:
・The introduction could be structured more logically by first describing MCA’s rarity, then summarizing existing molecular knowledge before introducing the study’s objectives.
・More citations are needed to support claims about MCA's clinical behavior.
4. Materials and Methods
Strengths:
・Describes tissue collection, sequencing, and analysis procedures in detail.
Suggested Improvements:
・Clearly define inclusion/exclusion criteria for case selection.
・Provide more details about the sequencing depth and coverage in WES analysis.
・Specify statistical tests applied to each type of data for clarity.
5. Results
Strengths:
・Presents a comprehensive analysis of clinical characteristics, sequencing findings, and BMP2 expression.
Suggested Improvements:
・Some descriptions of results (e.g., somatic mutation findings) could be presented more concisely.
・Ensure that all statistical results (e.g., p-values, effect sizes) are clearly reported.
6. Discussion
Strengths:
・Contextualizes findings within the existing literature.
・Explores potential clinical implications.
Suggested Improvements:
・The discussion should address limitations more explicitly, particularly the small sample size.
・Consider discussing whether BMP2 mutations have been reported in other cancer types.
・Address potential confounding factors in WES analysis.
・Please consider referring to the following article to expand the discussion. PMID: 39657258
7. Conclusion
Strengths:
・Summarizes key findings and potential implications.
Suggested Improvements:
・The conclusion should highlight future directions more explicitly (e.g., need for validation in larger cohorts, functional studies of BMP2).
・Avoid overgeneralizing findings given the small sample size.
8. References
Strengths:
・The reference list is comprehensive and relevant.
Suggested Improvements:
・Consider citing additional studies on BMP2’s role in cancer.
9. Figures and Tables
Strengths:
・Figures effectively illustrate key findings.
Suggested Improvements:
・Some figure legends could provide more explanatory context (e.g., defining key abbreviations used in the plots).

Reviewer 3 ·

Basic reporting

-

Experimental design

-

Validity of the findings

-

Additional comments

The authors describe three cases of a very rare subtype of breast carcinoma. In addition, they perform whole-exome sequencing finding mutations of the BMP2. The title accurately reflects the manuscript. The manuscript involves an important area of health. The manuscript is well written in terms of clarity, style, and use of English and has a logical construction. The discussion section explains the case in the context of published information. The conclusions accurately and clearly explain the main clinical message. The figures are of good quality and relevant to the clinical message. The references are appropriate and current.

---

## Round 0.2 · accepted · Accept

· Academic Editor

Accept

In evaluating the revised manuscript, I have carefully considered the reviewers’ comments, including their legitimate concerns regarding sample size, depth of interpretation, and methodological transparency, along with your point-by-point responses and the updated manuscript. While some inherent limitations remain typical of case-based molecular studies, I am confident that you have adequately addressed the key criticisms and revised the manuscript accordingly. Therefore, I believe your work is suitable for publication within its current scope.

Reviewer 2 ·

Basic reporting

The article  maintains a professional  tone through its  entire content written in  clear English. The literature references  together with background  information are adequate  for this study. The  research includes well-organized  figures and tables alongside  raw data. The  research results directly support the hypotheses through  strong evidence from  the study.

Experimental design

The research  presents an original  solution which effectively solves an essential knowledge  deficiency. The research followed both high technical and  ethical standards during  its execution. The research methods contain enough  detail to support replication efforts.

Validity of the findings

The  study provides important  findings although it does not directly measure  impact or novelty. The research data  show strong statistical validity  and effective control  measures. The research findings directly  support the study's  conclusions which match  the investigation's main question.

Additional comments

No further comments are  needed. The article  fulfills all requirements  of the journal.